# Fermented Whey Ewe’s Milk-Based Fruit Smoothies: Bio-Recycling and Enrichment of Phenolic Compounds and Improvement of Protein Digestibility and Antioxidant Activity

**DOI:** 10.3390/antiox12051091

**Published:** 2023-05-12

**Authors:** Ali Zein Alabiden Tlais, Elisabetta Trossolo, Stefano Tonini, Pasquale Filannino, Marco Gobbetti, Raffaella Di Cagno

**Affiliations:** 1Faculty of Science and Technology, Free University of Bolzano, 39100 Bolzano, Italy; altlais@unibz.it (A.Z.A.T.); etrossolo@unibz.it (E.T.); stonini@unibz.it (S.T.); marco.gobbetti@unibz.it (M.G.); 2Department of Soil, Plant and Food Science, University of Bari Aldo Moro, 70121 Bari, Italy; pasquale.filannino1@uniba.it

**Keywords:** whey ewe’s milk, smoothie, recycling, lactic acid bacteria, phenolic compounds, anthocyanins, antioxidant activity, protein digestibility, PDCAAS

## Abstract

This study aimed to recycle whey milk by-products (protein source) in fruit smoothies (phenolic compounds source) through started-assisted fermentation and delivering sustainable and healthy food formulations capable of providing nutrients that are unavailable due to an unbalanced diet or incorrect eating habits. Five lactic acid bacteria strains were selected as best starters for smoothie production based on the complementarity of pro-technological (kinetics of growth and acidification) traits, exopolysaccharides and phenolics release, and antioxidant activity enhancement. Compared to raw whey milk-based fruit smoothies (Raw_WFS), fermentation led to distinct profiles of sugars (glucose, fructose, mannitol, and sucrose), organic acids (lactic acid and acetic acid), ascorbic acid, phenolic compounds (gallic acid, 3-hydroxybenzoic acid, chlorogenic acid, hydrocaffeic acid, quercetin, epicatechin, procyanidin B2, and ellagic acid) and especially anthocyanins (cyanidin, delphinidin, malvidin, peonidin, petunidin 3-glucoside). Protein and phenolics interaction enhanced the release of anthocyanins, notably under the action of *Lactiplantibacillus plantarum*. The same bacterial strains outperformed other species in terms of protein digestibility and quality. With variations among starters culture, bio-converted metabolites were most likely responsible for the increase antioxidant scavenging capacity (DPPH, ABTS, and lipid peroxidation) and the modifications in organoleptic properties (aroma and flavor).

## 1. Introduction

The global fruit and vegetable consumption is below the World Health Organization (WHO) recommended threshold of at least 400 g of fruits and vegetables per day due to current lifestyle and eating habits [1]. This deficiency shifts the consumers orientations towards convenient ready-to-drink beverages [2], encouraging the food industry to develop new and alternative products such as smoothies with functional additives and interesting flavor combinations. Smoothies are non-alcoholic beverages which are typically semiliquid and made of mixed fruits, mainly berries, exotic, and citrus fruits. They are a great option to enhance the intake of bioactive compounds, particularly phenolic compounds, anthocyanins, and dietary fibers [3]. Nevertheless, phenolics are well known for their instability and low bioavailability, restricting the capability to exert their functionality [4]. Phenolics have also been shown to interact with proteins. This interaction can either negatively affect the release of phenolic compounds by generating a stable protein–phenolic complex or positively by catalyzing reactions that break down phenolic compounds and modify their chemical structure. This contradiction had been revealed through several studies [5,6,7,8]. Indeed, the influence of protein on the release of phenolic compounds is complex and context-dependent and requires further investigation. Therefore, the addition of a protein source to a phenolic-rich smoothie might be a promising strategy to affect the stability and bio-accessibility of phenolics and especially anthocyanins during consumption.

Whey milk is a protein-rich cheese processing waste. Several factors affect whey milk composition and quality, most notably the applied processing technology, which includes thermal processing and the protein precipitation method [9]. Whey milk represents an excellent source of lactoferrin, bovine serum albumin, lactoperoxidase, immunoglobulins, growth factors, bioactive peptides, and galacto-oligosaccharides [10]. Because of the inherent chemical composition, the consumption of isolated whey protein is encouraged to increase physical performance, modulate adiposity, and enhance immune functions in humans [11,12]. So far, only 50% of the whey milk produced worldwide is further processed and valorized for human consumption, despite the fact that whey production from cheese making exceeds 190 million tons per year [13]. Consequently, the disposal of whey remains a critical challenge due to its high biological oxygen demand (BOD) on the one hand and the wasted opportunity to generate profits from this abundant by-product on the other hand [14]. Recently, extensive studies have targeted the applicability of whey milk to design a novel beverage [15,16,17], supporting our approach to include whey milk in smoothie formulations.

Lactic acid fermentation, the eco-sustainable biotechnology, was proposed for recycling food wastes and converting them into a value-added product enriched in bioactive compounds [18,19]. Indeed, lactic acid bacteria (LAB) reveal a large portfolio of enzymes targeting various bioactive precursors during plant foods fermentation including phenolics which in turn result in various derivatives [20,21]. The proteolytic activity of LAB represents another key metabolism occurring during food fermentation which leads to an increase in protein digestibility [22,23,24]. To the best of our knowledge, no other studies are available about fermenting whey fruit-based products using LAB starters.

In this study, we aimed to investigate the metabolic behavior and technological properties of LAB strains isolated from different sources by usingwhey ewe’s milk-fruit juice as a growth model. Thereafter, the capability of best performing LAB to enrich the portfolio of phenolics and especially anthocyanins of whey milk fruit smoothies while also enhancing protein digestibility were highlighted. In a broad sense, we aimed to set up a biotechnological protocol to produce novel fruit smoothie formulations nourished with wasted whey milk and possessing high antioxidant properties.

## 2. Materials and Methods

### 2.1. The Plant Materials, Microorganisms, and Culture Conditions

Fruits (apple, strawberry, blueberry, raspberry, pear, and peach) were supplied by Zuegg Com (Lana, Bolzano, Italy). Whey cow, ewe, and goat milk were provided from Latteria di Lagundo (Lagundo, Bolzano, Italy). All samples were stored at 4 °C prior use. Fruit juices and puree were prepared as described later, and whey milk samples were stored at −20 °C for further analysis. Sixteen strains of LAB from the Culture Collection of the Department of Soil, Plant, and Food Science, University of Bari Aldo Moro (Bari, Italy), and the Micro4Food laboratory of the Faculty of Science and Technology, Free University of Bolzano (Bolzano, Italy) were used as starters in this study. To fully exploit the metabolic potential of LAB, the strains were chosen according to their environment of origin and ensuring a high degree of heterogeneity in terms of metabolism patterns. LAB strains were previously isolated from fruit and milk whey (data not published), with the exception of fructophilic lactic acid bacteria (FLAB) *Apilactobacillus kunkeei* BEE4, which was isolated from the gastro-intestinal tract of bees [25] (Appendix A). Cultures were maintained as stocks in 20% (v v^−1^) glycerol at −20 °C and routinely propagated at 30 °C for 24 h in MRS broth (Oxoid, Basingstoke, Hampshire, UK) for LAB and in FYP broth for FLAB.

### 2.2. Physico–Chemical and Biochemical Characterization

Total titratable acidity (TTA) was determined on 10 g of samples homogenized with 90 mL of distilled water using a Stomacher apparatus (Seward, London, UK), and expressed as the amount (mL) of 0.1 M NaOH to reach a pH of 8.3. The value of pH was measured by a Foodtrode electrode (Hamilton, Bonaduz, Switzerland). Chemical analyses concerning total proteins, fat content, and dry matter were determined by MilkoScan™ FT6000 (Foss Electric A/S, Hillerod, Denmark), based on Fourier-transformed infrared technology [26].

Carbohydrates were determined in water-soluble extract (W-SE) by HPLC analysis. Two grams of freeze-dried fruit powder were extracted with 20 mL of water/perchloric acid (95:5, *v*:*v*). In an ice bath, the mixture was sonicated (amplitude 60) with a macro-probe (Vibra-Cell sonicator; Sonic and Materials Inc., Danbury, CT, USA) for 1 min (two cycles, 30 s/cycle, 5 min interval between cycles). The suspension was stirred at room temperature for 1 h, kept at 4 °C overnight, and centrifuged for 10 min at 10,000 rpm. W-SE was filtered and stored at −20 °C until further use. A Spherisorb column (Waters, Milford, CT, USA) and a Perkin Elmer 200a refractive index detector were used to determine the concentrations of glucose, fructose, mannitol, and sucrose [25]. Carbohydrates standards were purchased from Sigma-Aldrich (Steinheim, Germany).

### 2.3. Starters Screening

Whey milk-based fruit juice was used as a growth model system to investigate the pro-technological performance of bacterial strains. Whey milk-based fruit juice medium was prepared as described by Filannino et al. [27] with a few modifications. Fruits (apple 10.5%, pear 17.5%, peach 10.5%, strawberry 2%, blueberry 1%, and raspberry 1%) were peeled, deseeded, and blended in distilled water 50% (*w*/*v*) and whey ewe’s milk (7.5%) (Classic Blender 400, PBI International). The percentage of the ingredients was defined based on several preliminary trials in which acceptable pH values (3.8–4.0) and texture were desired, whereas ewe’s milk was selected due to its superior sensory (pleasant taste and aroma) features and high protein content (1.02 g 100 g^−1^) compared to cow’s (0.73 g 100 g^−1^) and goat’s (0.78 g 100 g^−1^) whey milk. The mixture was shaken for 1 h at room temperature, centrifuged (10,000 rpm, 20 min at 4 °C), and sterilized by filtration through 0.22 µm membrane filters (Millipore Corporation, Bedford, MA, USA). Once prepared, whey fruit juice was kept at −20 °C until use. Cells of LAB and FLAB were harvested after 24 h by centrifugation (10,000 rpm, 10 min at 4 °C), washed twice in 50 mM sterile potassium phosphate buffer (pH 7.0), and singly inoculated in whey milk-based fruit juice to a final cell density of ca. 7.0 log CFU mL^−1^. Whey milk-based fruit juice was incubated at 30 °C, and the kinetics of growth were determined. Growth was monitored by measuring the optical density at 620 nm. Kinetics of growth was determined and modeled according to the Gompertz equation as modified by Zwietering et al. [28]: *y* = *k* + *A exp*{− *exp*[(*μ_max_e*/*A*)(*λ* – *t*) + 1]}, where *k* is the initial level of optical density (OD_620_ units), *A* is the difference in OD_620_ units between inoculation and the stationary phase, *μ_max_* is the maximum growth rate (OD_620_ units h^−1^), *λ* is the length of the lag phase (hours), and *t* is the time (hours). Data were fitted using a non-linear regression procedure of the Statistica 7.0 software (Statsoft, Tulsa, OK, USA). The acidification capacity of all strains was determined after 24 h of incubation at 30 °C. The pH was measured by a Foodtrode electrode (Hamilton, Bonaduz, Switzerland).

Further explorative screening for the ability to metabolize free phenolics was carried out through the total phenolic compounds assay according to the Folin-Ciocalteu method [29]. The data were expressed as mg gallic acid equivalent (GAE) per liter. The in vitro antioxidant activity was determined through the radical scavenging capacity on 1,1-diphenyl-2-picrylhydrazyl radical (DPPH) [30]. Aiming to investigate the synthesis of exo-polysaccharides (EPS), colonies from cell suspensions of each strain, pre-cultivated in MRS or FYP broth, were allowed to grow in MRS or FYP agar with the addition of 292 mM sucrose, 146 mM glucose, or 146 mM fructose. After incubation at 30 °C for 48 h, the synthesis of EPS was determined by the visual appearance of the mucoid colonies.

### 2.4. Fermentation of Whey Milk-Based Fruit Smoothies (WFS)

Fruits were previously washed with distilled water. Stems, skins, and woody endocarps (when needed) were removed, then fruits were homogenized through a vertical homogenizer (Treviso, Italy) at room temperature and mixed according to the following ratio: apple (21%), pear (35), peach (21), strawberry (4), blueberry (2), and raspberry (2). Blends were heated at 80 °C for 10 min and cooled at 25 °C before fortifying with 15% (*v*/*v*) of whey ewe’s milk. Subsequently, *Lactiplantibacillus plantarum* SL8 and BpL2, *Leuconostoc holzapfelii* PHE5, *Lactococcus lactis* WSL2, and *A. kunkeei* BEE4, selected as starters, were added at an initial cell density of ca. 7 Log CFU mL^−1^. WFS was fermented at 30 °C for 72 h. Samples were taken before and after fermentation. Raw WFS (Raw_WFS), without bacterial inoculum and incubation and WFS without inoculum but incubated at 30 °C for 72 h (Unstarted_WFS) were used as the controls. Values of pH, carbohydrates, and cell counts were determined as described above. Lactic and acetic acids were determined in W-SE by HPLC analysis equipped with an Aminex HPX-87H column (ion exclusion, Biorad) and a UV detector operating at 210 nm [19]. Organic acids standards were purchased from Sigma-Aldrich (Steinheim, Germany). With the exceptions of color, viscosity, and sensory analysis, the samples were freeze-dried (Epsilon 2-6D LSC plus freeze-drier, Martin Christ, Osterode am Harz, Germany) before being analyzed as described below.

### 2.5. Determination of Ascorbic Acid

The ascorbic acid content of all samples was determined according to Filannino et al. [31]. After extraction with a metaphosphoric acid solution, an aliquot of the extract was treated with D,L-dithiothreitol (Sigma Aldrich) to convert any dehydroascorbic acid present in the sample to ascorbic acid. The ascorbic acid content was determined by an HPLC system Ultimate 3000 (Dionex, Germering, Germany) equipped with a UV detector, column Ascentis RP Amide (250 mm × 4.6 mm; 5 μm) and column oven. Chromeleon Software vs. 7 (Dionex, Germering, Germany) was used to perform the analysis and to elaborate on the data. Solvents A (50 mM H_3_PO_4_, pH 3) and B (methanol) were isocratically eluted in 13 min for chromatographic separation. Twenty microliters of extract were injected, and elution was carried out at 25 °C with a flow rate of 1 mL min^−1^. The analyses of ascorbic acid were performed at UV wavelengths of 245 nm. The quantity of ascorbic acid was expressed as g 100 g^−1^ of dry weight.

### 2.6. Identification and Quantification of Free Phenolic Compounds

Aiming to understand the pattern of interaction between the whey protein and phenolic compounds, another control was required. Raw FS was made in the same way as Raw WFS but with water instead of whey milk. Phenolic compounds analyses were also carried out using methanol/water/hydrochloric acid (70:30:0.1, *v*:*v*) soluble extracts (MWH-SE) of samples. Freeze-dried materials (2 g) were homogenized with 20 mL of methanol/water/hydrochloric acid solution. In an ice bath, the mixture was sonicated (amplitude 60) with a macro-probe (Vibra-Cell sonicator; Sonic and Materials Inc., Danbury, CT) for 1 min (two cycles, 30 s/cycle, 5 min interval between cycles) in an ice-bath. The suspension was incubated at room temperature for 1 h under stirring conditions. The MWH-SE recovered by centrifugation (10,000 rpm for 10 min) was used after filtration. Targeted LC-MS/MS analysis of 45 free phenolic compounds was performed according to a revised version of the method previously designed and validated by Tlais et al. [18], by using an UHPLC Dionex 3000 (Thermo Fisher Scientific, Dreieich, Germany) equipped with a Waters Acquity HSS T3 column (1.8 μm, 100 mm × 2.1 mm) (Milford, MA, USA) and coupled to a TSQ Quantum™ Access MAX Triple Quadrupole Mass Spectrometer (Thermo Fisher Scientific, Germany) with an electrospray source. Target phenolics were detectable under multiple reaction monitoring (MRM) modes and the compounds were identified based on their reference standard, retention time, qualifier, and quantifier ion. The management of the chromatographic system and data acquisition was by Xcalibur software version 4.1 (Thermo Fisher Scientific, Germany).

### 2.7. Identification and Quantification of Anthocyanins

Anthocyanins were extracted from all samples as previously described by Barnes et al. [32]. Briefly, 5 mL of methanol/water/trifluoracetic acid soluble extracts (MWT-SE) (70:30:1, *v*:*v*:*v*) were added to 500 mg of freeze-dried powder. The mixture was vortexed at high speed and left undisturbed for 1 h. Then the mixture was sonicated for 20 min and then centrifuged at 2000× *g* for 20 min. The MWT-SE supernatant was filtered with a PTFE filter into an HPLC vial and stored at −80 °C until analysis. Separation, determination, and quantification of anthocyanins were performed by using the method validated for phenolic compounds [18] and described above with slight modifications in which targeted LC-MS/MS analysis of six anthocyanins was performed.

### 2.8. Antioxidant In Vitro Assays

Aiming to determine the radical scavenging capacity, W-SE and MWT-SE from samples were assessed. DPPH radical scavenging activity was measured by using the stable 1,1-diphenyl-2-picrylhydrazyl radical (DPPH) as reported above. ABTS radical scavenging activity was estimated by the Antioxidant Assay kit (Sigma-Aldrich) according to the manufacturer’s instructions. The principle of ABTS assay is the formation of a ferryl myoglobin radical from metmyoglobin and hydrogen peroxide, which oxidizes the ABTS (2,2′-azino-bis(3-ethylbenzothiazoline-6-sulphonic acid) to produce a radical cation, ABTS.^+^, a soluble chromogen that is green in color and can be determined spectrophotometrically at 405 nm. Trolox, a water-soluble vitamin E analog, was used as a control antioxidant. Lipid peroxidation was calculated using the Lipid Peroxidation Malondialdehyde (MDA) Assay Kit (Sigma-Aldrich) according to the manufacturer’s instructions. In this kit, lipid peroxidation was determined by the reaction of MDA with thiobarbituric acid (TBA) to form a colorimetric (532 nm) product, in proportion to the MDA present.

### 2.9. Amino Acids Profile

All samples were subjected to three separate hydrolysis sessions to obtain a complete and totally free individual amino acid profile. Acid hydrolysis based on AOAC Method 994.12 was performed to determine most amino acids, with few exceptions. For the determination of cysteine and methionine, a performic acid oxidation method (AOAC 994.12) was followed. Under the standard conditions of acid hydrolysis, tryptophan is unstable and cannot be analyzed effectively which necessitated alkaline sodium hydrolysis (AOAC 988.15). Total and individual amino acids were analyzed by a Biochrom 30 series Amino Acid Analyzer (Biochrom Ltd., Cambridge Science Park, England) with a Na-cation-exchange column (20 by 0.46 cm internal diameter). The limiting essential amino acid was identified as the one with the lowest value compared to FAO-recommended values for essential amino acids.

### 2.10. In Vitro Protein Digestibility (IVPD) and PDCAAS

The IVPD was evaluated using the Protein Digestibility Assay kit (Megazyme International, Wicklow, Ireland) according to the manufacturer’s instructions. PDCAAS was calculated by multiplying the amino acid score of the limiting amino acid and the IVPD.

### 2.11. Color, Viscosity, and Sensory Analysis

The color was measured using a CR-400 Chroma Meter. Samples were placed in petri dishes and filled to the top. The *L**, *a**, and *b** color space analysis method was used, where *L** represents lightness (white–black) and *a** and *b** corresponds to the chromaticity coordinates (red–green and yellow–blue, respectively). The dynamic viscosity was measured on ca. 20 g of samples at room temperature using a rotation viscosimeter (Anton Paar ViscoQC 300, Rivoli, Italy). The spindle was lowered inside the adapter containing the samples, and the viscosity readings were measured.

Sensory evaluation of all samples was carried out using Quantitative Descriptive Analysis (QDA) [33] by 10 non-trained panelists. After fermentation, samples were refrigerated, randomly coded, and served (10 g) at 15 °C together with non-salted table biscuits and still water. Samples were scored from 0 (lowest) to 10 (highest) for the following sensory attributes: taste, evaluated as sweetness, acidity, and astringency; appearance, referring to color intensity and absence of defects; texture, evaluated during scooping and mastication in the mouth; and flavor and aroma intensity.

### 2.12. Statistical Analysis

All analyses were carried out considering triplicates on three biological replicates. Data were submitted for analysis of variance by the General Linear Model (GLM) of the R statistical package (R, version 1.6.2, available at the rcompanion.org/handbook/ accessed on 11 January 2023). Pairwise comparison of treatment means was achieved by Tukey-adjusted comparison procedure with a *p*-value (*p*) < 0.05 [34]. Hierarchical clustering analysis, using the default method available in R and based on the Euclidean distance and the McQuitty linkage, was performed on data referring to starters screening parameters.

## 3. Results

### 3.1. Physico–Chemical Characterization of Raw Ingredients

The pH values of fruits ranged from 5.09 ± 0.00 to 2.97 ± 0.00 pH unit, with pear and raspberry having the highest and lowest values, respectively (Table 1). As expected, TTA values showed an opposite trend when compared to pH values. Fructose, glucose, mannitol, and sucrose were the most abundant sugars founds in all fruits with some variation depending on the fruit and the sugar (Table 1).

The physico–chemical composition of whey milks varied slightly based on the source of milk (Table 1). The goat’s whey milk showed the highest TTA (0.85 ± 0.1 mL of 0.1M NaOH) and fat content (1.18 g 100 g^−1^). Protein and dry matter were highly found in ewe’s whey milk (1.02 and 7.65 g 100 g^−1^, respectively).

### 3.2. Starters Screening

Sixteen strains of LAB were used to evaluate growth and acidification capacity in a whey milk-based fruit juice used as a model growth system. The inoculum cell density was ca. 7.0 Log CFU mL^−1^, and the initial value of pH was 3.88 ± 0.05. Based on the kinetic growth, the 16 strains exhibited various growth patterns during the incubation period (Figure 1A). The cell density (*A*) increased up to 0.08–0.32 OD_620_ units. Values of *µ_max_* and *λ* were in the range of 0.006–0.036 OD_620_ units h^− 1^ and 0.80–19.29 h, respectively. Based on these data, the less-performing strains were *Lc. lactis* WSL2 and *Leuc. mesenteroides* WCL1 (Figure 1A). Except for a few strains of *Leuc. mesenteroides* and *Leuc. pseudomesenteroides*, most of the strains caused a significant (*p* < 0.05) reduction in pH values after 24 h of incubation (Figure 1B).

To further screen the pro-technological traits of strains, total phenolic contents and radical scavenging activity of whey milk-based fruit juice were evaluated (Figure 1C). Under the condition of our study, the capability to modify the profile of phenolics was species- or strain-dependent. Compared to the raw sample (223.1 ± 5.91 mg L^−1^), the concentration of total phenols increased only for eight strains, mainly belonging to *L. plantarum* (three strains), *Lc. Lactis* (1), *Leuc. pseudomesenteroides* (2) and *Leuc. mesenteroides* (2). The highest release was found for *L. plantarum* SL8 (301.0 ± 64.55 mg L^−1^). The increase in total phenolic is reflected on the DPPH scavenging activity. Visual inspection of colonies distinguished *L. plantarum* (three strains) and *Leuc. mesenteroides* (3), *A. kunkeei* (1) and *Luec. holzapfelii* as efficient EPS producers due to the large and dense mucus colonies on agar media. Based on growth, acidification, total phenols, DPPH radical scavenging activity, and EPS production strains were clustered in six groups (A–F) (Figure 1D). Cluster A contained only *Leuc. mesenteroides* WCL1, which had the lowest growth and acidifying capacity, and consequently, was excluded from our selection. Representative strains from each of the five remaining clusters, comprising *L. plantarum* SL8 and BpL2, *Leuc. holzapfelii* PHE5, *Lc. lactic* WSL2, and *A. kunkeei* BEE4, were selected as potential starters for making whey milk-based fruits smoothies.

### 3.3. Fermentation of Whey Milk-Based Fruit Smoothies

The protocol for the processing of whey milk-based fruit smoothies was set up. Selected starters were single inoculated at an initial cell density of ca. 7.0 log CFU mL^−1^. The fermentation was carried out at 30 °C for 72 h. Whey milk-based fruit smoothies (WFS) without bacterial inoculum and incubated under the same conditions were used as the control (unstarted). Samples were taken before and after fermentation.

The cell density of presumptive LAB in Raw_WFS was 3.89 ± 0.09 log CFU mL^−1^. Before fermentation, the main difference between started and unstarted was regarding the cell numbers of lactic acid bacteria. Due to the inoculum, Started_WFS had cell densities of lactic acid bacteria ca. 10,000 times higher than Unstarted_WFS. After 72 h of fermentation, cell numbers of presumptive lactic acid bacteria of Started_WFS increased to 7.51–8.82 log CFU mL^−1^. Presumptive lactic acid bacteria of Unstarted_WFS reached a cell density of 6.73 ± 0.13 Log CFU mL^−1^. When WFS was started with *Leuc. holzapfelii* PHE5 and *L. plantarum* BpL2, the lowest and highest increase in cell density were found, respectively. The LAB growth mirrored the differences in acidification capacity. Compared to Raw_WFS (3.81 ± 0.01), almost all Started_WFS showed lower and significant (*p* < 0.05) pH values. The highest (*p* < 0.05) pH reduction was found in WFS started with *L. plantarum* BpL2 (3.23 ± 0.01).

### 3.4. Analysis of Sugars, Organic Acids, and Vitamin C

Fructose (403.4 ± 8.76 mg g^−1^ DW) and glucose (188.4 ± 5.62 mg g^−1^ DW) were the main sugars detected in Raw_WFS, followed by mannitol (51.7 ± 13.07 mg g^−1^ DW), lactose (65.5 ± 3.07 mg g^−1^ DW), and sucrose (10.00 ± 1.69 mg g^−1^ DW) (Table 2 and Appendix A). After 72 h of incubation, fructose was significantly (*p* < 0.05) consumed only in WFS fermented with *Lc lactis* WSL2 (320.13 ± 4.90 mg g^−1^ DW), *Leuc. holzapfelii* PHE5 (326.56 ± 7.96 mg g^−1^ DW), *A. kunkeei* BEE4 (329.85 ± 9.89 mg g^−1^ DW), and *L. plantarum* BpL2 (349.29 ± 7.69 mg g^−1^ DW). The highest (*p* < 0.05) reduction of glucose was found in WSL2_WFS (149.9 ± 0.59 mg g^−1^ DW) and BEE4_WFS (152.5 ± 3.65 mg g^−1^ DW). Mannitol was found at the highest (*p* < 0.05) level in BEE4_WFS (151.4 ± 3.62 mg g^−1^ DW), PHE5_WFS (148.0 ± 2.45 mg g^−1^ DW), and WSL2_WFS (128.2 ± 1.87 mg g^−1^ DW). All selected strains fully consumed sucrose, whereas the level of lactose was unaffected by the incubation. The main microbial metabolites were lactic and acetic acids (Table 2 and Appendix A). The highest (*p* < 0.05) level of lactic acid was observed in BpL2_WFS (75.3 ± 2.84 mg g^−1^ DW), followed by SL8_WFS (46.6 ± 0.04 mg g^−1^ DW) and BEE4_WFS (41.0 ± 1.76 mg g^−1^ DW), whereas Unstarted_WFS (11.8 ± 0.61 mg g^−1^ DW) had the lowest. Acetic acid values among fermented samples ranged from 0.39 ± 0.05 to 3.62 ± 0.22 mg g^−1^ DW.

Compared to Raw_WFS (1.3 ± 0.06 g 100 g^−1^ DW), the vitamin C content measured as ascorbic acid significantly (*p* < 0.05) decreased in Unstarted_WFS (1.06 ± 0.07 g 100g^−1^ DW) (Table 2 and Appendix A), whereas all selected starters had enhancing or preserving effects (*p* > 0.05).

### 3.5. Free Phenolic Compounds

Aiming to evaluate the effects of protein addition and fermentations on free phenolic compounds, their profile was analyzed through LC-ESI-MS/MS analysis. The highest peaks of 15 phenolic compounds belonging to various chemical classes were identifiable using external standards (Table 3 and Appendix A, Appendix A). When compared to Raw_FS, the addition of whey protein enhanced significantly (*p* < 0.05) the release of five phenolic compounds (gallic acid, chlorogenic acid, isoquercetin, epicatechin, and ellagic acid) (Appendix A). Chlorogenic acid was the most prevalent phenolic compound (62.9 ± 0.06 µg g^−1^ DW) in WFS_Raw, followed by procyanidin B2 (25.4 ± 0.51 µg g^−1^ DW) and epicatechin (25.0 ± 0.49 µg g^−1^ DW). Other identified phenolics ranged from 9.4 ± 0.20 µg g^−1^ DW (for *p*-coumaric acid) to 0.9 ± 0.22 µg g^−1^ DW (for quercetin). The use of selected starters substantially altered the profile of phenolic compounds compared to WFS_Raw and WFS_Unstarted, and the effect was species and strain dependent. Gallic acid significantly (*p* < 0.05) increased in WSL2_WFS (0.45 ± 0.01 µg g^−1^ DW) and BEE4_WFS (0.43 ± 0.01 µg g^−1^ DW), whereas the lowest concentration was detected in BpL2_WFS (0.24 ± 0.00 µg g^−1^ DW). *L. plantarum* SL8 caused a high reduction of 3-hydroxybenzoic acid, chlorogenic acid, procyanidin B2, and ellagic acid, whereas the opposite trend was shown by *Leuc. holzapfelii* PHE5 for 3-hydroxybenzoic acid, by *A. kunkeei* BEE4 for chlorogenic acid, by *Leuc. holzapfelii* PHE5 and *Lc. lactis* WSL2 for procyanidin B2, and by only *Lc. lactis* WSL2 for ellagic acid. Hydrocaffeic acid, which was not found in Raw_WFS appeared only following fermentation with *L. plantarum* BbL2. Unlike Raw_WFS, quercetin was not detectable in PHE5_WFS. *L. plantarum* BpL2 and SL8 led to the highest (*p* < 0.05) decrease in epicatechin amounts. Other compounds, including *p*-coumaric acid, isorhamnetin, naringenin, phloridzin, isoquercetin, phloretin, and vanillin, did not exhibit a significant difference (*p* > 0.05) among all samples.

Anthocyanins, cyanidin, malvidin, peonidin, petunidin 3-glucoside, and delphinidin were identified in raw and Started_WFS (Figure 2 and Appendix A). Only malvidin showed significantly higher values in Raw_WFS compared to Raw_FS (Appendix A). Petunidin 3-glucoside was the major anthocyanin compound (77.90 ± 15.08 µg g^−1^ DW) in Raw_WFS, followed by delphinidin (11.51 ± 1.93 µg g^−1^ DW), and malvidin (6.29 ± 0.15 µg g^−1^ DW), whereas cyanidin and peonidin were the least abundant (5.11 ± 1.64 and 2.97 ± 0.24 µg g^−1^ DW, respectively). Although cyanidin and delphinidin markedly increased in almost Started_WFS, *L. plantarum* BpL2 led to their highest value equal to 100.9 ± 7.82 µg g^−1^ DW and 155.75 ± 15.02 µg g^−1^ DW, respectively. Malvidin and peonidin showed almost the same trend but to a lesser extent among the Started_WFS. Except for *Leuc. holzapfelii* PHE5, all the other selected starters caused a significant (*p* < 0.05) increase in petunidin 3-glucoside.

### 3.6. Antioxidant Activity

To achieve high reliability, antioxidant capacity measurements employing ABTS, DPPH, and lipid peroxidation assays were evaluated (Figure 3). A remarkable increase (*p* < 0.05) of the antioxidant activity was found in W-SE of samples fermented with *L. plantarum* BpL2 (53.1 ± 4.67 mM g^−1^ DW) with the ABTS assay compared to Raw_WFS (43.0 ± 1.38 mM g^−1^ DW) and Unstarted_WFS (41.8 ± 8.04 mM g^−1^ DW), whereas no substantial differences were appreciable among different samples using MWT-SE. Likewise, DPPH radical scavenging of W-SE of BpL2_WFS (30.2 ± 0.08 mmol BHT g^−1^ DW) followed by PHE5_WFS (29.9 ± 1.06 mmol BHT g^−1^ DW) showed significantly higher antioxidant activity than Raw_WFS (25.5 ± 0.09 mmol BHT g^−1^ DW) and Unstarted_WFS (26.6± 0.21 mmol BHT g^−1^ DW). When using MWT-SE, fermented WFS did not show any significant difference compared to the raw sample. Regardless of the type of extracts, WFS fermented *with L. plantarum* BpL2 induced the lowest lipid peroxidation (1209.8 ± 0.36 nmol g^−1^ DW for W-SE and 1336.7 ± 21.40 nmol g^−1^ DW for MWT-SE).

### 3.7. Amino Acids Profile, Protein Digestibility, and PDCAAS

The effect of fermentation on the protein digestibility of whey fruit smoothies was notable, even though no significant (*p* > 0.05) differences in protein content were found among the started samples compared to the raw one, which ranged from 0.62 ± 0.03 g 100 g^−1^ DW (Raw_WFS) to 0.75 ± 0.1 g 100 g^−1^ DW (BEE4_WFS) (Table 4). In order to evaluate the total free amino acids, raw and started whey milk-based fruit smoothies were subjected separately to acid, performic oxidation, and alkaline hydrolysis. Based on the amino acids profile, high variations were found among the samples (Appendix A). The variations in Lys, His, Pro, and Arg reflected mostly the variation in the protein digestibility among the samples. The lowest Lys value and the highest concentration of His, Pro, and Arg were associated with the highest significant (*p* < 0.05) protein digestibility found in whey milk-based fruit smoothies started with *L. plantarum* BpL2 (87.3 ± 0.00%). The lowest significant (*p* < 0.05) protein digestibility was found when *A. kunkeei* BEE4 (79.8 ± 0.02%) and *Lc. lactis* WSL2 (79.5 ± 0.03%) were used as starters. According to essential amino acids values recommended by FAO, amino acids score and, consequently, first limiting amino acid were determined. Except for the sample started with *L. plantarum* SL8, where Val had the lowest ratio, Ile was identified as the first limiting amino acid in all samples. Amino acids score and PDCAAS increased from 0.46 and 0.38 to 0.6 and 0.52, respectively, for BpL2_WFS, and it was reduced from 0.46 and 0.38 to 0.31 and 0.25, respectively, for WSL2_WFS (Table 4).

### 3.8. Color, Texture, and Sensory Analysis

Color lightness (*L**) substantially (*p* > 0.05) differed among samples, showing the highest values in Unstarted_WFS, WSL2_WFS, and BEE4_WFS and the lowest values in Raw_WFS_Raw and BpL2_WFS (Table 5). The scaler quantity of red-green (*a**) had the highest significant values in BpL2_WFS (24.25 ± 0.75), followed by Raw_WFS (21.79 ± 0.03), and then the other samples. On the contrary, the yellow–blue (*b**) index of Raw_WFS (12.63 ± 0.23) and followed by BpL2_WFS (14.4 ± 0.60) was significantly (*p* < 0.05) lower than other samples, with Unstarted_WFS showing the highest values. The texture, measured as viscosity, ranged from 70.5 ± 3.40 to 94.3 ± 3.62, with no significant (*p* > 0.05) differences among the samples. On a 10-point scale, seven sensory attributes were evaluated: appearance, texture, odor, and taste. The score of appearance, texture, sweetness, and astringency did not significantly (*p* > 0.05) differ among samples. The highest score of aroma was found in WFS_Raw (7.2 ± 1.46) and WFS started with *L. plantarum* SL8 (7.2 ± 1.32) and BpL2 (6.8 ± 1.46). An almost similar trend of scores was found for flavor distinguishing WFS_Raw and WFS_WSL2. The highest acidity score was found when whey milk-based fruit smoothies were inoculated with *L. plantarum* BpL2.

## 4. Discussion

Human health has always been a major concern of international associations and food industries, which establish guidelines and develop new food formulations with the aim of promoting lifelong health [35]. The inadequate consumption of basic nourishment, especially represented by fruit and vegetable intake [36] motivates these institutions towards exploring sustainable food solutions that ensure nutritional needs. In this scenario, we proposed a suitable biotechnological framework to formulate a novel fruit smoothie nourished with whey milk and fermented with diverse lactic acid bacteria.

Commonly, mango and banana are the most popular fruits for smoothie manufacture. In this study, other phenolic aptitudes were employed such as apple, strawberry, peach, pear, blueberry strawberry, and raspberry [37]. Despite the poor sensory quality, the rich protein and peptide contents justified the enrichment of smoothie formulations with whey milk not only to bio-recycle but also to enhance phenolic compounds release [38,39]. Indeed, the porous structure of the protein can trap phenolics and change their availability for absorption [40]. Recently, whey fortification in food formulations is becoming very attractive for the consumers, such as bread [41], cereals blends [42], baby food [43], and fermented infant formula [44].

Based on their metabolic potentiality, several species of autochthonous LAB, previously isolated from the same fruits and whey milk by-products were chosen. *Leuconostoc holzapfelii*, *Leuconostoc mesenteroides*, and *Lactiplantibacillus plantarum* were the only species isolated previously from these fruits [45]. Due to additional parameters such as the source, composition, and applied dairy processing technology, a higher diversity of LAB was found in whey milk samples [9]. In fact, *Lactococcus lactis* is mainly used in the fermentation of various dairy products and is also known for having probiotic properties, which modulate the gut microbiome functionality [46]. To meet our goal, several starter selection criteria were investigated, with a focus on environmental adaptation as well as other potential metabolic traits (EPS release, phenolic compounds release, and DPPH scavenging activities) [2,18,19]. Consequently, *L. plantarum* BpL2, *L. plantarum* SL8, *Leuc. holzapfelii* PHE5, *Lc. lactis* WSL2 autochthonous strains, and allochthonous *A. kunkeei* BEE4 were selected as best performing starters to ferment whey ewe’s milk-based fruit smoothies. In accordance with previous studies, *L. plantarum* isolated from most of the food matrices showed the highest adaptability to harsh environments mediated by gene mechanisms of regulation and adaptation [21]. Adopted mechanisms of adaptation reflected various species and strain traits, which in turn explain the variation in sugar and organic acid metabolism throughout the fermentation. Sucrose was the preferred substrate of all selected LAB which is hydrolyzed into glucose and fructose [47]. On the contrary and inconsistent with previous studies, lactose was not metabolized by any of the selected starters [48]. Fructose utilization by *Lc. lactis* WSL2, *Leuc. holzapfelii* PHE5 and especially *A. kunkeei* BEE4 as alternative external electron acceptor explained the mannitol production [49]. As the main microbial metabolite and in line with previous studies, the formation of lactic acid by LAB was strain dependent and favored mainly by *L. plantarum* [50].

Afterward, the effect of lactic fermentation on bioactive compounds was highlighted through different analyses. Ascorbic acid is the most well-known antioxidant and is essential in the human diet [51]. Based on previous studies, the effect of fermentation on vitamin C content was contradictory. For instance, the ascorbic acid was reduced during cabbage fermentation since it was used as a precursor in ascorbigen pathway formation [52]. On the other side and in accordance with our findings, lactic fermentation can have modulating and preserving effect on ascorbic acid content, suggesting the role of pH reduction to prevent ascorbate auto-oxidation when the redox potential changes [31].

Phenolics, as essential elements of plant-based products, have been explored as fermentation substrates for better components bioavailability and enhanced functional properties [53]. In our study, we highlighted the capacity of *Leuc. holzapfelii, Lc. Lactis*, and *A. kunkeei* to modify the profile of phenolic compounds, species that have not commonly been considered for this purpose. The whey milk-based fruit smoothie, as expected, showed a rich phenolic profile composed of nineteen compounds, including the anthocyanins. In line with previous studies, species and strain-specific phenolic metabolic features of LAB starters during whey fruit smoothie fermentation were confirmed [54]. Based on our findings, only *Lc. lactis* and *A. kunkeei* induced gallic acid release, which might be attributed to the presence of tannin acyl hydrolase acting on hydrolysable tannins to release glucose and gallic acid. Hence, gallate decarboxylase involvement in subsequent transformation of gallic acid into pyrogallol might explain their low values of gallic acids. Other phenolic acids (3-hydroxybenzoic and chlorogenic acids) were hydrolyzed by *L. plantarum* SL8 and liberated by *Leuc. holzapfelii* (3-hydroxybenzoic acid) and *A. kunkeei* (chlorogenic acid). The same trend was found for epicatechin and procyanidin B2. Hydrolysis of chlorogenic acid into free phenolic acids and release of flavonoids and flavanols were related to esterase activity [31,55,56]. The biotransformation of epicatechin and procyanidin B2 by *L. plantarum* was previously found during pomegranate fermentation into new phenol derivatives [57]. Under the same fermentation condition, the release of ellagic acid from ellagitannins or ellagic acid-glycosides was speculated [58] but contradicts our findings that revealed its hydrolysis. Production of dihydrocaffeic acid exclusively by one strain of *L. plantarum* suggested phenolic acid reductase activity toward caffeic acid resulting from the degradation of chlorogenic acid [59].

Anthocyanins are other phenolic compounds belonging to the flavonoid group. Broadly, anthocyanins are characterized by their limited bioavailability and instability [60,61]. To achieve their functions on health, these flaws need to be addressed during food processing and before consumption. Indeed, processing is expected to negatively affect the concentration of flavonoids and anthocyanins. Paradoxically, LAB starters boosted the anthocyanin content in started whey milk-based fruit smoothies. The addition of a protein source to a fruit-based smoothie greatly influenced the stability and reactivity of anthocyanins, especially during fermentation, when whey proteins are denatured by the high temperature, which eventually increases the interaction of anthocyanins with whey proteins and pectin [62]. Notably, *L. plantarum* BpL2 showed the highest release of free anthocyanins, reinforcing the hypothesis that glycosylated flavonoids (e.g., cyanidin-3-O-glucoside, malvidin-3-O-glucoside) were hydrolysis by β-glucosidase activity [20]. The opposite trend for petunidin 3-glucoside contradicts this argument but might be explained by various enzymes liberated during fermentation that are capable of breaking down plant cell walls, making such compounds more accessible [63].

Extensive research employing various substrates and assays to evaluate antioxidant activity failed to reveal predictable changes in antioxidant activity throughout fermentation. Nevertheless, plenty of studies, including ours, have concluded that started assisted fermentation can maintain or enhance the antioxidant activity when compared to unstarted samples [27,64]. By considering a panel of extracts and assays, we assume that modifications in bioactive composition especially anthocyanins point out an increase in antioxidant activity after fermentation. The highest release of anthocyanins by *L. plantarum* BpL2 was associated with the strongest inhibition of lipid peroxidation. Because antioxidant defense mechanisms are primarily generated against free radicals such as ABTS and DPPH, our fermentations rigidified these systems by enhancing ABTS and DPPH scavenging activity, especially when water-soluble extracts were used. This contradicted our expectations about phenolics being more soluble in methanol water-soluble extracts, but it did highlight the role of mannitol, ascorbic acid, or other water-soluble protein–phenolic complexes generated upon protein addition as reducing agents, free-radical scavengers, and singlet oxygen quenchers, confirming our previous assumption [65]. Furthermore, this assumption does not conflict the fact that some phenolic compounds such as epicatechin and procyanidin B2, are soluble in water and can act as antioxidants [66].

The nutritional value of proteins is initiated only after it has been enzymatically or chemically transformed into small peptides or amino acids during food (bio)processing [67]. Hence, PDCAAS and protein digestibility are crucial features to evaluate protein quality and to estimate the protein availability for intestinal absorption after digestion, and consequently its utilization, respectively. Guo, [68] mentioned that a score of 1.00 indicates a perfect PDCAAS. By a long history of investigations, the majority of fruits appeared as marginal protein sources [69,70]. Whey milk showed a low PDCAAS value (0.38), comparable to that of some whey protein supplements previously mentioned in other studies [71,72]. Several studies have documented the efficacy of LAB to increase the digestibility of whey milk protein through their proteolytic system [73,74], which is partially consistent with our findings. Under our experimental conditions, only *L. plantarum* strains showed a significant positive effect on protein digestibility. The different proteolysis degrees of whey protein by LAB might also explain the discrepancies in PDCAAS values among fermented whey milk-based fruit smoothies, which were calculated based on protein digestibility and according to FAO-suggested values for essential amino acids. Likewise, only *L. plantarum* strains revealed an increase in PDCAAS values compared to the raw and unstarted samples. Based on the available literature, the PDCAAS value of whey protein is contradictory. Overall, the incorporation of protein from whey by-product distinguishes our smoothie formulations from those available on the market as solely phenolic sources, dietary fibers, vitamins, and minerals [3]. However, being fermented raised this advantage when offering products with lower sugar content but higher nutritional and functional features. Lowering sugar, in particular, has the potential to reduce microbiota associated with an obese phenotype [75], whereas anthocyanins can help to alleviate allergy symptoms by suppressing the production of pro-inflammatory cytokines [76]. It is noteworthy that our fermented formulations have the potential to contribute to several of the Sustainable Development Goals of the United Nations 2030 Agenda by promoting waste recycling and providing good health and well-being [77].

Finally, sensory evaluation such as color and texture are critical when assessing fermented smoothies [78]. Selected LAB starters affect the color of whey milk-based fruit smoothies in a diverse manner. The high release of anthocyanins as natural pigments by *L. plantarum* strains might justify the preserving of color features. When compared to unfermented samples, fermented matrices showed differences mainly in terms of aroma and flavor. Basically, the distinctive sensory features of fermented samples were primarily the result of the different profiles of sugars, organic acids, phenolic compounds, anthocyanins, and amino acids.

## 5. Conclusions

Research is increasingly oriented towards the development of innovative, healthy, and sustainable foods for humans, capable of providing nutrients that compensate for the lack of essential elements due to an unbalanced diet or incorrect eating habits. Our study established a biotechnological protocol to produce novel fermented fruit smoothies nourished with protein from whey ewe’s milk. While we confirmed the higher capability of *L. plantarum* compared to other species to adapt and interact in a model system rich in phenolics, we also speculated that phenolic–protein interactions may improve the phenolics’ bioactivity and release during fermentation on one side, and protein digestibility on the other side. Consequently, our findings offered a further referenced role of sugars, organic acids, amino acids, phenolic compounds, and especially anthocyanins metabolism not only in potentiating the antioxidant characteristics but also preserving organoleptic properties in sustainable whey-fruit-based formulations and offering more advantages over unfermented and heat-treated fruit smoothies commonly available to consumers.

## Figures and Tables

**Figure 1 antioxidants-12-01091-f001:**
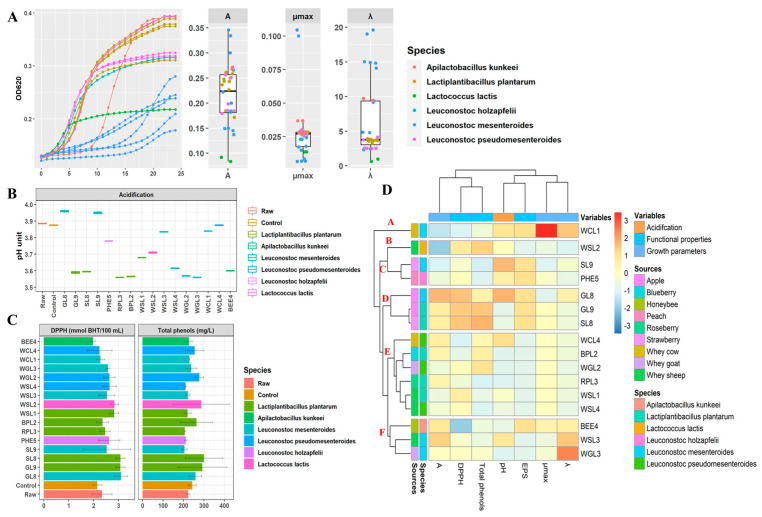
Parameters of the kinetics of growth (**A**) and acidification (**B**) of lactic acid bacteria during fermentation of whey milk-based fruit juice at 30 °C for 24 h and total free phenolic compounds (mg/L) and DPPH radical scavenging capacity (mmol BHT/100 mL) (**C**). Pseudo-heat map showing growth parameters, acidification, total free phenolic compounds, and DPPH radical scavenging capacity during fermentation of whey milk-based fruit juice at 30 °C for 24 h (**D**). Rows are clustered using Euclidean distance and McQuitty linkage. The color scale shows the differences between the standardized data.

**Figure 2 antioxidants-12-01091-f002:**
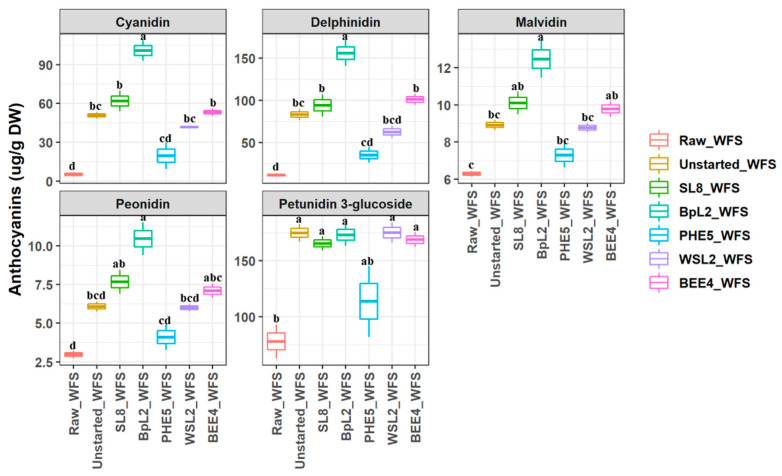
Quantification of anthocyanins compounds (µg g^−1^ DW) by LC-ESI-MS/MS in methanol/water/trifluoracetic acid soluble extract (MWT-SE) obtained from raw whey milk-based fruit smoothiesw (Raw_WFS), WFS without microbial inoculum (Unstarted_WFS), and Started_ WFS, which were incubated for 72 h at 30 °C. Fermentation (Started_WFS) was with selected single cultures of *Lactiplantibacillus plantarum* SL8 and BpL2, *Leuconostoc holzapfelii* PHE5, *Lactococcus lactic* WSL2 and *Apilactobacillus kunkeei* BEE4. Bars with different superscript letters differ significantly (*p* < 0.05).

**Figure 3 antioxidants-12-01091-f003:**
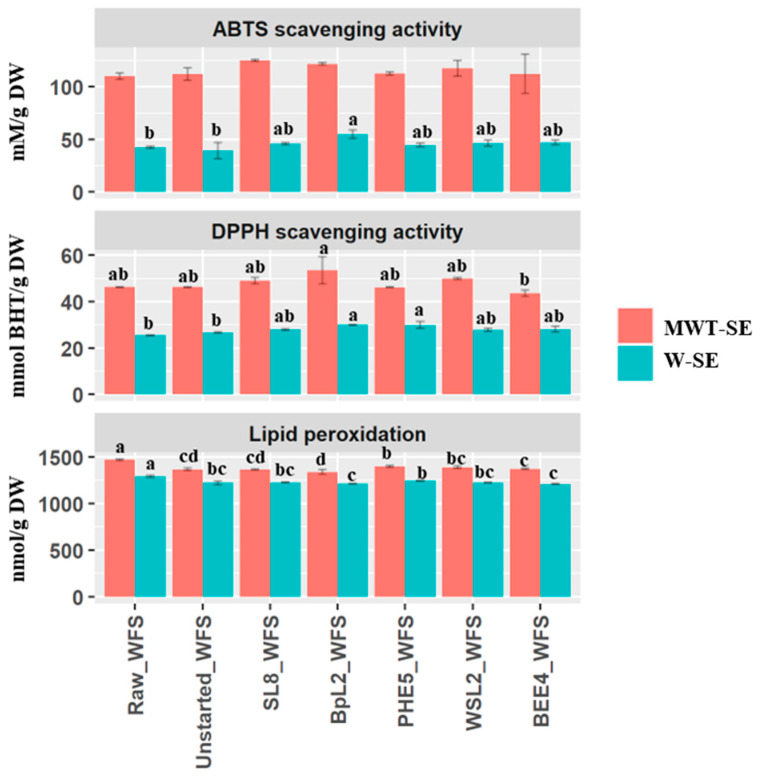
ABTS^.^ (mM g^−1^ DW) and DPPH^.^ (mmol BHT g^−1^ DW) radical scavenging activities, and lipid peroxidation (nmol g^−1^ DW) of water-soluble extract (W-SE) and methanol/water/trifluoracetic acid soluble extract (MWT-SE) obtained from raw whey milk-based fruit smoothie (Raw_WFS), WFS without microbial inoculum (Unstarted_WFS), and Started_WFS, which were incubated for 72 h at 30 °C. Fermentation (Started_WFS) was with selected single cultures of *Lactiplantibacillus plantarum* SL8 and BpL2, *Leuconostoc holzapfelii* PHE5, *Lactococcus lactic* WSL2, and *Apilactobacillus kunkeei* BEE4. Bars with different superscript letters differ significantly (*p* < 0.05).

**Table 1 antioxidants-12-01091-t001:** Values of pH, total titratable acidity (mL of 0.1 M NaOH), and concentration (mg g^−1^ DW) of carbohydrates and organic acids in fruit matrices.

Fruits	pH	TTA (mL of 0.1 M NaOH)	Sugars (mg g^−1^ DW)			
	pH	TTA	Fructose	Glucose	Mannitol	Sucrose
**Apple**	4.07 ± 0.1	4.95 ± 0.2	604.69 ± 4.1	179.64 ± 29.2	11.89 ± 1.3	6.03 ± 0.9
**Strawberry**	3.41 ± 0.0	13.9 ± 0.3	221.88 ± 28.5	301.88 ± 13.6	0.00	0.00
**Pear**	5.09 ± 0.0	4.2 ± 0.1	457.59 ± 37.2	88.07 ± 16.2	149.38 ± 11.6	2.97 ± 0.9
**Peach**	3.79 ± 0.0	5.65 ± 0.1	222.10 ± 5.0	312.37 ± 13.6	0.00	20.67 ± 2.1
**Raspberry**	2.97 ± 0.0	43.15 ± 0.4	159.63 ± 6.8	175.78 ± 2.5	0.00	0.00
**Blueberry**	3.08 ± 0.0	18.55 ± 0.2	177.58 ± 22.8	191.00 ± 23.0	0.00	0.00

**Table 2 antioxidants-12-01091-t002:** Sugars (mg g^−1^ DW), organic acids (mg g^−1^ DW) and ascorbic acid (g 100 g^−1^ DW) quantification in raw whey-fruit smoothie (Raw_WFS), WFS without microbial inoculum (Unstarted_WFS), and Started_WFS, which were incubated for 72 h at 30 °C. Fermentation (Started_WFS) was with selected single cultures of *Lactiplantibacillus plantarum* SL8 (SL8_WFS) and BpL2 (BpL2_WFS), *Leuconostoc holzapfelii* PHE5 (PHE5_WFS), *Lactococcus lactic* WSL2 (WSL2_WFS), and *Apilactobacillus kunkeei BEE4* (BEE4_WFS).

Samples	Fructose	Glucose	Mannitol	Sucrose	Lactose	Lactic Acid	Acetic Acid	Ascorbic Acid
**Raw_WFS**	403.39 ± 8.76 ^a^	188.39 ± 5.62 ^a^	51.72 ± 13.07 ^b^	10.00 ± 1.69 ^a^	65.52 ± 3.07	4.10 ± 0.25 ^d^	0.21 ± 0.05 ^c^	1.30 ± 0.06 ^a^
**Unstarted_ WFS**	375.68 ± 8.23 ^ab^	174.78 ± 8.81 ^ab^	79.20 ± 1.41 ^b^	0 ± 0 ^b^	58.58 ± 1.99	11.82 ± 0.61 ^cd^	2.31 ± 0.20 ^b^	1.06 ± 0.07 ^b^
**SL8_WFS**	394.00 ± 4.24 ^a^	191.84 ± 2.64 ^a^	74.38 ± 0.99 ^b^	0 ± 0 ^b^	60.92 ± 1.49	46.65 ± 0.04 ^b^	0.78 ± 0.07 ^c^	1.39 ± 0.21 ^a^
**BpL2_WFS**	349.29 ± 7.69 ^bc^	175.93 ± 5.01 ^ab^	62.91 ± 1.31 ^b^	0 ± 0 ^b^	69.37 ± 1.67	75.34 ± 2.84 ^a^	0.39 ± 0.05 ^c^	1.36 ± 0.07 ^a^
**PHE5_WFS**	326.56 ± 7.96 ^c^	165.12 ± 3.23 ^ab^	148.00 ± 2.45 ^a^	0 ± 0 ^b^	66.91 ± 3.10	20.39 ± 2.29 ^c^	3.06 ± 0.43 ^ab^	1.28 ± 0.02 ^a^
**WSL2_WFS**	320.13 ± 4.9 ^c^	149.86 ± 0.59 ^b^	128.22 ± 1.87 ^a^	0 ± 0 ^b^	66.01 ± 0.37	24.65 ± 2.25 ^d^	0.81± 0.10 ^c^	1.20 ± 0.05 ^a^
**BEE4_WFS**	329.85 ± 9.89 ^c^	152.46 ± 3.65 ^b^	151.42 ± 3.62 ^a^	0 ± 0 ^b^	60.76 ± 0.69	40.99 ± 1.76 ^b^	3.62 ± 0.22 ^a^	1.21 ± 0.03 ^a^

^a–d^ Means within the column with different letters are significantly different (*p* < 0.05).

**Table 3 antioxidants-12-01091-t003:** Quantification of phenolic compounds (µg g^−1^ DW) by LC-ESI-MS/MS in methanol/water/ hydrochloric acid soluble extract (MWH-SE) obtained from raw whey-fruit smoothie (Raw_WFS), WFS without microbial inoculum (Unstarted_WFS), and Started_WFS, which were incubated for 72 h at 30 °C. Fermentation (Started_WFS) was with selected single cultures of *Lactiplantibacillus plantarum* SL8 (SL8_WFS) and BpL2 (BpL2_WFS), *Leuconostoc holzapfelii* PHE5 (PHE5_WFS), *Lactococcus lactic* WSL2 (WSL2_WFS), and *Apilactobacillus kunkeei* BEE4 (BEE4_WFS).

Compounds	Raw_WFS	Unstarted_WFS	SL8_WFS	BpL2_WFS	PHE5_WFS	WSL2_WFS	BEE4_WFS
**Gallic acid**	0.31 ± 0.00 ^bc^	0.35 ± 0.01 ^b^	0.31 ± 0.01 ^c^	0.24 ± 0.00 ^d^	0.27 ± 0.01 ^cd^	0.45 ± 0.01 ^a^	0.43 ± 0.01 ^a^
**3- hydroxybenzoic acid**	13.31 ± 1.31 ^ab^	19.54 ± 3.97 ^ab^	8.64 ± 0.68 ^b^	12.56 ± 2.50 ^ab^	23.72 ± 2.84 ^a^	15.87 ± 0.50 ^ab^	18.83 ± 0.53 ^ab^
**Chlorogenic acid**	62.94 ± 0.06 ^ab^	69.36 ± 6.02 ^a^	44.97 ± 0.69 ^b^	63.38 ± 1.51 ^ab^	66.00 ± 6.10 ^ab^	68.94 ± 3.98 ^ab^	72.06 ± 6.02 ^a^
** *p* ** **-coumaric acid**	9.38 ± 0.20	9.82 ± 0.26	9.24 ± 0.02	9.21 ± 0.17	9.44 ± 0.07	9.73 ± 0.09	9.7 ± 0.26
**Hydrocaffeic acid**	0 ± 0 ^b^	0 ± 0 ^b^	0 ± 0 ^b^	0.99 ± 0.45 ^a^	0 ± 0 ^b^	0 ± 0 ^b^	0 ± 0 ^b^
**Phloridzin**	11.23 ± 3.24	10.24 ± 4.00	11.10 ± 2.10	12.52 ± 2.77	10.61 ± 2.05	9.65 ± 3.92	11.97 ± 3.26
**Isorhamnetin**	5.59 ± 1.16	5.98 ± 0.11	4.62 ± 0.03	6.25 ± 0.42	4.49 ± 0.02	6.29 ± 0.46	6.25 ± 0.20
**Naringenin**	1.46 ± 0.45	1.73 ± 0.11	1.18 ± 0.03	1.82 ± 0.13	1.47 ± 0.06	1.74 ± 0.02	1.84 ± 0.01
**Phloretin**	6.14 ± 0.04	6.06 ± 0.01	6.03 ± 0.00	6.15 ± 0.04	6.05 ± 0.01	6.09 ± 0.02	6.07 ± 0.01
**Quercetin**	0.92 ± 0.22 ^ab^	0.23 ± 0.03 ^bc^	0.80 ± 0.09 ^ab^	1.059 ± 0.14 ^a^	0 ± 0 ^b^	0.96 ± 0.14 ^a^	0.94 ± 0.08 ^ab^
**Isoquercetin**	3.59 ± 0.13	2.55 ± 0.92	3.83 ± 0.17	4.23 ± 0.34	3.31 ± 0.63	3.03 ± 0.94	2.52 ± 0.17
**Epicatechin**	25.04 ± 0.49 ^b^	24.19 ± 1.00 ^b^	11.62 ± 0.94 ^c^	6.70 ± 0.76 ^d^	33.51 ± 0.47 ^a^	26.77 ± 0.24 ^b^	25.41 ± 0.64 ^b^
**Procyanidin B2**	25.45 ± 0.51 ^ab^	23.96 ± 0.20 ^ab^	19.85 ± 0.23 ^b^	26.35 ± 3.70 ^ab^	31.72 ± 0.03 ^a^	31.54 ± 3.95 ^a^	25.62 ± 0.39 ^ab^
**Ellagic acid**	8.70 ± 0.78 ^ab^	7.91 ± 0.93 ^ab^	3.32 ± 0.04 ^b^	7.74 ± 1.38 ^ab^	5.44 ± 0.10 ^ab^	9.79 ± 1.06 ^a^	8.69 ± 1.47 ^ab^
**Vanillin**	6.13 ± 0.12	6.22 ± 0.01	6.06 ± 0.00	6.20 ± 0.09	6.21 ± 0.08	6.13 ± 0.02	6.17 ± 0.04

^a–d^ Means within the row with different letters are significantly different (*p* < 0.05).

**Table 4 antioxidants-12-01091-t004:** Protein digestibility and protein digestibility corrected amino acids score (PDCAAS) of raw whey-fruit smoothie (Raw_WFS), WFS without microbial inoculum (Unstarted_WFS), and Started_WFS, which were incubated for 72 h at 30 °C. Fermentation (Started_WFS) was with selected single cultures of *Lactiplantibacillus plantarum* SL8 (SL8_WFS) and BpL2 (BpL2_WFS), *Leuconostoc holzapfelii* PHE5 (PHE5_WFS), *Lactococcus lactic* WSL2 (WSL2_WFS), and *Apilactobacillus kunkeei* BEE4 (BEE4_WFS).

Amino Acids	FAO * (mg/g Protein)	Raw_WFS	Unstarted_WFS	SL8_WFS	BpL2_WFS	PHE5_WFS	WSL2_WFS	BEE4_WFS
**Thr**	34	76.5	69.6	67.7	71.9	70.6	54.8	74.8
**Val**	35	23.8	19.7	17.6	21.4	16.5	16.3	18
**Cys + Met**	25	137.7	150.9	130.8	160.7	114.6	100.8	117.7
**Ile**	28	12.9	13.9	15.8	16.8	9.8	8.7	9.9
**Leu**	66	47.4	38.2	44.1	55.5	41.6	37	45.3
**Tyr + Phe**	63	77.6	56.8	60.2	85.2	65.6	78.2	77.6
**Lys**	58	52.2	44.4	39.2	37.9	44.7	49.2	56
**His**	19	24.8	20.6	17.7	24.8	19.5	18.4	20.2
**Trp**	11	79.8	70	70.9	77.1	70.7	69	69.8
**Protein (g/100 g DW)**		0.62 ± 0.03	0.7 ± 0.02	0.71 ± 0.05	0.68 ± 0.03	0.67 ± 0.01	0.73 ± 0.01	0.75 ± 0.1
**Protein digestibility (%)**		82.6 ± 0 ^c^	82.85 ± 0.02 ^c^	84.22 ± 0.01 ^b^	87.35 ± 0 ^a^	81.75 ± 0.04 ^c^	79.5 ± 0.03 ^d^	79.85 ± 0.02 ^d^
**First limiting Amino acid**		Ile	Ile	Val	Ile	Ile	Ile	Ile
**Amino acids score**		0.46	0.5	0.5	0.6	0.35	0.31	0.35
**PDCAAS**		0.38	0.41	0.42	0.52	0.29	0.25	0.28

^a–d^ Means within the column with different letters are significantly different (*p* < 0.05). * FAO recommended values for essential amino acids.

**Table 5 antioxidants-12-01091-t005:** Color indices (*L**, *a**, and *b**, white–black, red–green, and yellow–blue coordinates, respectively), texture (viscosity), and sensory properties of raw whey-fruit smoothie (Raw_WFS), WFS without microbial inoculum (Unstarted_WFS), and Started_WFS, which were incubated for 72 h at 30 °C. Fermentation (Started_WFS) was with selected single cultures of *Lactiplantibacillus plantarum* SL8 (SL8_WFS) and BpL2 (BpL2_WFS), *Leuconostoc holzapfelii* PHE5 (PHE5_WFS), *Lactococcus lactic* WSL2 (WSL2_WFS) and *Apilactobacillus kunkeei* BEE4 (BEE4_WFS).

Samples	Color indices			Texture	Sensory Properties						
	*L**	*a**	*b**	Viscosity	Appearance	Texture	Aroma	Flavor	Acidity	Sweetness	Astringency
**Raw_WFS**	31.99 ± 0.55 ^c^	21.79 ± 0.03 ^b^	12.63 ± 0.23 ^d^	94.31 ± 3.62	7.2 ± 0.74	6.8 ± 0.97	7.2 ± 1.46 ^a^	7.2 ± 1.46 ^a^	4.2 ± 2.31 ^ab^	5.4 ± 1.85	3.4 ± 2.93
**Unstarted_WFS**	39.63 ± 0.63 ^a^	16.35 ± 0.34 ^d^	22.04 ± 0.54 ^a^	80.94 ± 6.5	5.2 ± 1.93	6.4 ± 1.35	6.2 ± 0.74 ^ab^	5.8 ± 0.97 ^ab^	4.8 ± 1.46 ^ab^	5.2 ± 1.16	3.2 ± 2.71
**SL8_WFS**	35.65 ± 0.35 ^b^	19.5 ± 0.50 ^c^	17.38 ± 0.63 ^c^	75.95 ± 6.01	6.4 ± 1.02	6.4 ± 1.01	7.2 ± 1.32 ^a^	6.4 ± 1.01 ^ab^	5.8 ± 2.13 ^ab^	5.4 ± 1.01	4.0 ± 3.34
**BpL2_WFS**	32.19 ± 0.19 ^c^	24.25 ± 0.75 ^a^	14.4 ± 0.60 ^d^	93.25 ± 1.42	7.2 ± 0.74	6.2 ± 1.60	6.8 ± 1.46 ^a^	5.0 ± 1.67 ^ab^	7.4 ± 0.48 ^a^	5.4 ± 1.01	6.0 ± 3.16
**PHE5_WFS**	37.36 ± 0.83 ^ab^	14.83 ± 0.18 ^d^	18.12 ± 0.38 ^bc^	73.85 ± 7.61	5.2 ± 2.40	6.0 ± 1.41	5.2 ± 1.16 ^ab^	5.2 ± 2.31 ^ab^	3.0 ± 2.19 ^b^	5.0 ± 2.09	2.8 ± 2.48
**WSL2_WFS**	39.20 ± 0.60 ^a^	15.96 ± 0.04 ^d^	21.14 ± 0.41 ^a^	80.98 ± 6.47	4.6 ± 1.85	5.8 ± 1.16	3.4 ± 1.62 ^b^	3.4 ± 1.35 ^b^	3.4 ± 2.41 ^ab^	4.6 ± 0.80	4.2 ± 2.40
**BEE4_WFS**	38.65 ± 0.25 ^a^	18.55 ± 0.15 ^c^	20.36 ± 0.36 ^ab^	70.51 ± 3.40	5.8 ± 0.97	6.4 ± 0.80	5.6 ± 1.01 ^ab^	6.0 ± 0.89 ^ab^	5.0 ± 1.26 ^ab^	5.6 ± 1.01	4.8 ± 2.63

^a–d^ Means within the columns with different letters are significantly different (*p* < 0.05).

## Data Availability

Not applicable.

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
