# Peer review of "Fermented Whey Ewe’s Milk-Based Fruit Smoothies: Bio-Recycling and Enrichment of Phenolic Compounds and Improvement of Protein Digestibility and Antioxidant Activity"

_antioxidants, 2023, doi:10.3390/antiox12051091_

Round 1
Reviewer 1 Report
Manuscript ID: antioxidants-2368266
The manuscript entitled “Fermented whey ewe’s milk-based fruit smoothies: bio-recycling and enrichment of phenolic compounds, and improvement of protein digestibility and antioxidant activity” concerns an integrated approach to investigate the behavior of lactic acid bacteria strains isolated from different sources by using a whey ewe’s milk-fruit juice as growth model. The capability of selected lactic acid bacteria to enrich of obtained products such as whey milk-fruit smoothies by higher concentration of phenolics and especially anthocyanins was highlighted. A suitable framework of biotechnological design of novel fruit smoothie formulations nourished with whey milk and possessing high antioxidant properties was proposed by Authors.
Considering the above, I note that the subject of the work is suitable for the publication in the Antioxidants. But some significant data are missing here and the lack of the criticism is the is the main problem. More details should be specified, including the comparison between proposed products and others usually available for consumers. Maybe it's worth thinking about and discussing the problem of lactose intolerance and other issues related to allergies.
So, I recommend the minor revision according to specific comments.
Remarks:
In abstract section data on the qualitative composition of obtained products, should be presented.
Introduction part - the originality and novelty of work are questionable. It could be explained in this section.
Figure 1 contains too many details that are not quite clear, please allow to zoom in of this part.
Identified compounds should be indicated on chromatograms as supplementary materials.
Conclusions do not indicate elements of scientific novelty or a new approach to the studied problem.
Summary:
The current version of the manuscript still needs minor revision and after an including of a completion it is recommended for the publication in the Antioxidants.
Reviewer 2 Report
The authors should clearly formulate the aims of their study in the abstract as well as at the end of the introduction.
Is there any preliminary evaluation of the LAB starter strains available? Regarding their safety, technological properties or probiotic potential?
In lines 610-611, the authors state that "When compared to raw sample, fermented matrices showed differences mainly in terms of aroma and flavor."
-First, the authors are comparing non-fermented with fermented samples. All samples are raw, in the sense that they were not cooked.
-Second, what about safety? Did the authors evaluate the safety and maybe also the shelf-life of fermented vs. non-fermented fruit smoothies?
Did the authors evaluate the presence of live bacteria in the fermented smoothies? And the contribution towards a better health of these potentially probiotic bacteria?
The authors could further highlight the contribution of these fermented whey ewe’s milk-based fruit smoothies towards achieving the Sustainable Development Goals of the United Nations 2030 Agenda.
